# Mesenchymal Stromal Cells as a Driver of Inflammaging

**DOI:** 10.3390/ijms24076372

**Published:** 2023-03-28

**Authors:** Svetlana Lyamina, Denis Baranovskii, Ekaterina Kozhevnikova, Tatiana Ivanova, Sergey Kalish, Timur Sadekov, Ilya Klabukov, Igor Maev, Vadim Govorun

**Affiliations:** 1Molecular Pathology of Digestion Laboratory, A.I. Yevdokimov Moscow State University of Medicine and Dentistry, Delegatskaya Str., 20/1, 127473 Moscow, Russia; 2Scientific Research Institute for Systems Biology and Medicine, Nauchniy Proezd, 18, 117246 Moscow, Russia; 3Research and Educational Resource Center for Cellular Technologies, Peoples’ Friendship University of Russia (RUDN University), 117198 Moscow, Russia; 4National Medical Research Radiological Centre of the Ministry of Health of the Russian Federation, 249036 Obninsk, Russia

**Keywords:** MSC-secretome, immunomodulatory, SASP, pro-inflammatory cytokines

## Abstract

Life expectancy and age-related diseases burden increased significantly over the past few decades. Age-related conditions are commonly discussed in a very limited paradigm of depleted cellular proliferation and maturation with exponential accumulation of senescent cells. However, most recent evidence showed that the majority of age-associated ailments, i.e., diabetes mellitus, cardiovascular diseases and neurodegeneration. These diseases are closely associated with tissue nonspecific inflammation triggered and controlled by mesenchymal stromal cell secretion. Mesenchymal stromal cells (MSCs) are known as the most common type of cells for therapeutic approaches in clinical practice. Side effects and complications of MSC-based treatments increased interest in the MSCs secretome as an alternative concept for validation tests in regenerative medicine. The most recent data also proposed it as an ideal tool for cell-free regenerative therapy and tissue engineering. However, senescent MSCs secretome was shown to hold the role of ‘key-driver’ in inflammaging. We aimed to review the immunomodulatory effects of the MSCs-secretome during cell senescence and provide eventual insight into the interpretation of its beneficial biological actions in inflammaging-associated diseases.

## 1. Introduction

Life expectancy has risen significantly over the past decades [1]. Obviously aging impacts human health and increases risk of diabetes, cardiovascular, autoimmune and neurodegenerative diseases. Age-related conditions are tightly associated with chronic nonspecific inflammation and accumulation of senescent cells in multiple tissues [2]. Cellular and molecular aspects of aging have become a central issue for longevity research groups. Much of the literature in previous decades discussed aging and longevity predominantly in a limited paradigm of cellular senescence and chronic diseases, while most recent studies enhanced discussion of inflammaging phenomenon [3]. Among the various definitions of inflammaging, the most accepted one characterizes it as ‘chronic, sterile, low-grade inflammation’ that accompanies aging as disturbances in sophisticated balance between pro- and anti-inflammatory cell signaling and responses [4,5]. Novel data on mesenchymal stem/stromal cells (MSCs) cross-talks via secretome in the context of inflammation have a certain clinical potential for the treatment of age-related diseases.

Adult stem cells including MSCs are the most commonly used cell type for regenerative medicine and tissue engineering since the first clinical trial in 1995 [6,7]. Since the mid1990s a considerable amount of literature postulated cellular differentiation as a key-issue of their action, while recent studies emphasize the role of immunomodulatory cross-talks derived by MSCs secretome [8].

MSCs are commonly identified according to the criteria of the Mesenchymal and Tissue Stem Cell Committee of the International Society for Cellular Therapy [9] The minimal criteria include positive labeling for CD105, CD73 и CD90 (more than 95% of cells) and negative for CD45, CD34, CD14 or CD11b, CD79a or CD19 and HLA-DR. Importantly, MSC subpopulations presumably differ in CD105 expression, depending on cell source, cell culture conditions and passage [10,11]. Furthermore, several pieces of evidence suggest the association of CD 105 with the immunomodulatory properties of MSCs [12,13].

Currently, MSCs-therapy spread as an effective tool for a range of autoimmune and immune-mediated inflammatory diseases [14]. MSCs were found to exhibit a unique immunoregulatory effect on immune cells, i.e., T- and B-lymphocytes, NK-cells, macrophages and dendritic cells [15]. Since 2000, both experimental and clinical studies have shown the effectiveness of MSC transplantation for graft-versus-host disease (GVHD), systemic lupus erythematosus (SLE), rheumatoid arthritis (RA), and inflammatory bowel disease (IBD) [15].

A growing list of clinical trials accumulated numerous reports on adverse events and side effects of cell-based therapies caused by the proinflammatory profile of MSC-secretome [16]. Being presented in various tissues, MSCs are both regulators and actors of local regeneration and immunomodulation [17]. MSC continuously secrete large extracellular vesicles, microvesicles, and exosomes containing a cocktail of cell-signaling molecules well-known under the complex definition of secretome [4]. Proliferating studies explained therapeutic effects of MSCs within the concept of secretome-associated cellular cross-talks [18]. Thus, MSC-secretome could be either a part therapeutic mechanism or a key trigger for following tissue alteration. The hypothesis is further supported by observed changes in MSC-secretome in a variety of diseases including immune- and age-related conditions [19].

The secretome is less immunogenic compared to MSCs and could be an ideal tool for cell-free regenerative therapies and even tissue engineering [20]. Engineered MSC secretome could be integrated in the framework of nanomedicine as a delivery system, enabling precision effects on certain tissue dysfunctions. The most recent results of MSC secretome as cell-free treatment have shown positive impact on inflammatory and age-related diseases [21]. In particular, the effect was shown in skeletal muscles regeneration (acute and chronic muscle injuries, atrophic muscle diseases) [18], intervertebral discs traumatic and degenerative changes [22], various dermatological disorders [23], liver failure [24], cardiovascular and neurodegenerative diseases—Alzheimer’s and Parkinson’s diseases [21], osteoarthritis [25]. MSC secretome also has a context-dependent dual role: the secretome triggers senescence of other MSCs and contributes to the key pathologic stages in ischemic organ changes and disorders (incl. smooth muscle cells calcification, skeletal muscles degeneration, organ failure, pancreatic degeneration) [26,27,28].

Currently clinical trials with secretome-based tests are at the very early stages of planning and development. In our review, we aimed to discuss the recent evidence of associations between MSCs secretome and age-related changes in tissue homeostasis via the paradigm of inflammaging. We also reviewed the relevance of secretome-based tests as an upcoming diagnostic tool for regenerative medicine.

## 2. Senescence of MSCs—Possible Inducers and Triggers

MSCs at late passages could be characterized as a heterogeneous population with a high proportion of cells in a state of irreversible proliferative arrest. Unlike apoptotic cells, those cells remain metabolically active, accumulate continuously, and are able to manifest their effects via the microenvironment to other cells [29].

Aging of MSC is always associated with morphological changes (enlarged and flattened shape), loss of proliferative and differentiative capabilities, qualitative and quantitative changes (intracellular organelles and secretome) [30]. Those aged cells are commonly described as senescent cells. Being able to maintain their functional activity for a sufficiently long period, they continuously interact with the microenvironment causing local and systemic effects [31,32,33,34,35]. Obviously, senescence is both a physiologically and pathologically relevant program.

Cell response to various external and internal stimuli can lead to senescence induction, premature aging and inflammaging-related conditions. Changes include progressive shortening of telomeres, mitogenic signals, oncogenic activation, oxidative and genotoxic stress, epigenetic changes, chromatin disorganization, impaired proteostasis, mitochondrial dysfunction [2]. Different types of signals cause different types of senescence: telomere-dependent replicative senescence, programmed senescence, premature senescence caused by non-telomeric stress (incl. oncogene-induced senescence), senescence induced by unresolved DNA damage, epigenetically induced senescence and mitochondrial dysfunction associated senescence [36]. So cellular senescence is primarily classified into replicative, oncogene-induced, stress-induced and developmental types [2]. MSCs in culture demonstrated terminal ability to proliferate before entering a phase of replication arrest [37]. In general, the MSCs’ life span in vitro ranges from 30 to 40 population doublings [38]. Senescence impairs MSC function both in vitro and in vivo. In particular, MSCs from older donors had downscaled proliferation and differentiation capacities compared to MSCs from younger donors [39].

Despite numerous distinctive phenotypic features, the identification of senescent cells remains difficult, since the majority of the markers could be also expressed by cells under certain specific conditions, incl. acute tissue damage and high lysosomal activity.

Identification of cellular senescence is commonly simplified to the detection of cell cycle arrest. Senescent studies are mainly focused on proteins p16, p21, p53, miR-146a microRNA and decreased expression of phosphorylated retinoblastoma protein (pRB). Analysis of nuclear protein Ki67 could be also considered as a suitable tool to judge about senescent condition. Ki67 is expressed during all active phases of the cell cycle, while its lower expression reflects decreased proliferative activity [2]. Senescent cells are also characterized by enhanced activity of lysosomal hydrolases. In particular, β-galactosidase is known as Senescence-associated β-galactosidase (SA-β-Gal). The major macromolecular changes involve the cell nucleus as well. Changed organization of heterochromatin and structural proteins is typical for senescent cells. Thus, the list of commonly used markers could be extended with Senescence-Associated Heterochromatin Foci (SAHF), heterochromatin protein 1 (HP1), H3K9me3, histone H2A variant of macroH2A and lamin B1 protein [40].

## 3. Immunomodulatory Effect of MSCs

Immunomodulatory effects of MSCs distinguish them from other stem cell types. Multiple preclinical trials have also shown that MSCs-based therapy is an effective way to treat refractory inflammatory diseases (incl. autoimmune diseases, acute GVHD, etc.) [41,42]. Currently the mechanism of the MSCs immunosuppressive properties is still shadowed. Increasing evidence indicates that MSCs can develop immunosuppressive properties by regulating the functional phenotypes and activities of innate and adaptive immune cells. MSCs modulate the microenvironment of immune cells and transform their functional phenotype: T cells shifted into a tolerant Treg phenotype; macrophages transformed into an immunoregulatory M2-macrophages phenotype; NK-cells transformed into non-free functional status. MSCs also suppress the synthesis of immunoglobulins and increase the replication of regulatory B-cells (Bregs) [43,44,45,46]. Distinct immunomodulatory mechanisms could be classified into cell-to-cell contact dependent and independent ways of immunosuppression [47,48,49]. In addition to soluble factors, MSCs realize immunosuppression effects via surface molecules. Direct cell-to-cell communication via *Gal-9* and *PD-L1* of AD-MSCs was found more powerful to trigger T-cell apoptosis than indirect induction [47]. Furthermore, contact communication is also accompanied by the secretion of anti-inflammation cytokines. Probably, direct cellular contacts can be a synergic effect that triggers the MSCs secretome and promotes immune tolerance.

Several lines of evidence suggest that expression of MSCs profile markers (CD14, CD34, CD44, CD73, and HLA-DR) are not related with cell differentiation and their initial origin or cell source. However, CD105 was shown to correlate with immunomodulation capacity [50]. The subpopulation of CD105 negative mesenchymal stem cells showed enhanced immunomodulation capacity compared to CD105 positive mesenchymal stem cells. Immunomodulation of CD105-negative MSCs may relate to their autocrine production of TGF-β1 [50]. Traditionally, it has been argued that MSCs can differ in CD105 expression being differentiated into osteoblasts, chondroblasts, and adipocytes [51]. Interestingly, several publications described the increased CD105 expression as typical for aged MSCs [50,51,52,53].

The immunomodulatory potential of MSCs is also significantly reduced during healthy aging. MSCs from young donors have enhanced anti-inflammatory properties that are widely used in regenerative medicine. In contrast, studies have shown that “old” MSCs tend to exhibit pro-inflammatory features, while their immunosuppressive capacity is impaired [30,54]. The decrease in MSCs immunosuppressive properties was confirmed in the comparative study with different donor cells at rising passages: early passage MSCs with the least immunosuppressive ability were more effective than late passage cells with the highest immunosuppression ability [55]. These findings suggest a correlation between a pro-inflammatory MSCs signature with donor age. BM-MSCs donated by adults in different ages (13–80 years) had different IL-6 concentrations associated with the age [56]. AD-MSC in elderly patients (over 65 years old) with atherosclerosis showed a tendency to release the pro-inflammatory secretome. The finding was further confirmed by increased levels of IL-6, IL-8 and monocyte chemoattractant protein 1 (MCP1) followed by decreased ability to suppress proliferation and activation of T-cells [57].

## 4. MSCs Secretome as Immunomodulatory Modality

In the previous section, we mentioned that differentiation, aging and senescence of MSCs are mediated by cellular cross-talks via growth factors and inflammatory cytokines. In particular, young MSCs could retire into cellular senescence in response to *transforming growth factor-β* (TGFβ2), while inhibition of TGF-β receptors has been found to promote proliferation of undifferentiated MSCs [58]. Furthermore, anti-TGFβ antibodies can initiate the regression of their aging processes [59].

Senescent cell is capable of regulating the basic processes itself incl. proliferation, migration, differentiation, and tissue remodeling [60]. Cell niche self-remodeling is a complex effect caused by a cocktail of bio-active molecules secreted into the extracellular space. Subsequently secreted factors could be assumed as SASP-senescence-associated secretory phenotype [61,62].

The SASP composition and strength depends on the inducer and duration of senescence, microenvironment and cell type [55].

SASP-family includes cytokines, chemokines, growth factors, metalloproteinases, extracellular vesicles [Figure 1]. There is a great variability in SASP composition and strength depending on the inducer and duration of senescence, environment and cell type [62]. So, the apparent diversity of substances complicates finding unique signatures [2]. The observed outcomes are both context dependent and cell type specific. However, NF-κB dependent pro-inflammatory factors are the key components of SASP with IL-6 and IL-8 being the most conserved and robustly expressed cytokines [2].

In radiation-induced senescent models BM-MSCs in vitro increased their expression of IL-6 with reduced immunosuppressive potential. The findings were confirmed in further experiments in vivo, when murine MSCs after irradiation also lost their protective immunosuppressive function [2].

Almost all the known senescence-inducing stimuli (incl. DNA damage, dysfunctional telomeres, genomic damage, epigenomic perturbation, mitogenic proliferative signals, oxidative stress, etc.) lead to prolonged DNA-damage response expressing SASP to different extents [21]. In contrast, SASP is not detectable in cells where senescence is induced by expression *p21WAF1/CIP1* or *p16INK4A* [2]. Therefore, DNA damage is an essential driver of SASP. However, the study by Freund et al. (2011) revealed a novel canonical DNA damage response signaling that regulates SASP independently via p38MAPK [63]. He demonstrated that p38MAPK induced SASP mainly upregulating NF-κB activity. Similarly, the induction of senescence by mitochondrial dysfunction presents a distinct secretory phenotype [64].

The Atlas of SASP-factors was presented in 2020; however, it failed to summarize the universal SASP signatures. Senescence associated secretory phenotype is nonspecific, context dependent and highly heterogeneous which can be regulated at multiple different levels. The difficulty in the identification of a general regulatory mechanisms restricts its utility as an unequivocal marker for senescence as no unique form of SASP is known to exist [65]. Nevertheless, SASP can be used as an additional biomarker of the cellular aging phenotype [66].

SASP includes cytokines, including pro-inflammatory cytokines (IL-1, IL-6, and IL-8), chemokines, such as monocyte and leukocyte chemotactic proteins (MIP1a, MCP2, CXCL9, and CXCL10), growth factors, metalloproteinases, extracellular vesicles. All these factors affect both MSCs and immune cells, as well as the extracellular matrix, to create an inflammatory microenvironment.

In particular, interleukins, IFN-γ, MCP-1 and MMP2 are present in enormous amounts in the secretome of senescent cells. The systematic understanding of how SASP interacts with the immune system is still lacking. Traditionally, immunomodulatory effects of MSCs were described in the paradigm of down-regulation of T-cell proliferative activity and impairment of dendritic cells via suppression of MHC II [67,68,69]. However, recent evidence suggests that SASP of aged MSCs recruits immune cells, including macrophages, natural killer cells, neutrophils, and T-lymphocytes to eliminate senescent cells [70]. At the same time, SASP can also exhibit an immunosuppressive function itself [2,71]. In particular, TLR-4 in an inflammatory environment stimulates MSCs to acquire an immunosuppressive phenotype [17]. Thus, normally MSCs demonstrate anti-inflammatory functions, while senescent cells have clear pro-inflammatory effects due to SASP. Clinicians continue debate about the safety and best strategies for cell-therapy for elderly patients. SASP secretome is considered to explain ineffectiveness and adverse events caused in elderly patients [72]. Recently, scientists have examined the effects of aged MSCs on hematopoietic stem cells. Secreted SASPs were shown to activate the inflammatory transcriptome in hematopoietic cells impairing their functionality [73].

Immunomodulatory functions are known to be associated with the expression of CD105 on MSCs. Thus, CD105-negative MSCs have a significant inhibitory effect on the expression of lymphocytes in comparison with CD105-positive cells. Probably, the expression of CD105 on MSCs may be part of a complex response to TGF-b. Indeed, CD105 is a component of the TGF-b receptor. Multiple studies have evaluated the role of TGF-b in immunomodulation [73,74]. A correlation between CD105 and immunomodulatory cell properties, as well as possible competition of CD105 with TGF-b produced by MSCs, has been already suggested. MSCs produce TGF-b1 and part of that molecule can bind to the TGF-b receptor, while CD105 is its component. A lack of CD105 expression leads to insufficient TGF-b1 effect affecting T cells [50]. CD105-positive MSCs have been shown to exhibit TGF-beta/SMAD2 signaling in mice [75]. The feedback effect of TGF-b1 on CD105-positive MSCs inhibits the production of proinflammatory iNOS/NO and IL-6 [50]. Furthermore, young MSCs could retire into cellular senescence in response to transforming growth factor-β (TGFβ2), while inhibition of TGF-β receptor has been found to promote proliferation of undifferentiated MSCs [58]. Furthermore, anti-TGFβ antibodies can initiate the regression of their aging processes [76].

## 5. Immunomodulation and Inflammaging via MSC and MSCs Secretome

SASP components of aging MSCs are both: markers and triggers of aging. Both roles were clearly confirmed during assessment of MSCs derived from human endometrium [77]. Indeed, SASP of BM- and AD-MSCs was mainly characterized by a signaling ability to maintain and induce aging in their niche [78]. Interestingly, the mechanism regulating SASP in MSCs has been attributed to GATA4 mediating MCP-1 expression in progerin- and/or prelamin-dependent pathways [79]. The secretion may be also induced by depletion of Hedgehog signaling pathway (IHH) ligands during aging of bone marrow MSCs. This type includes activation of IL-6, indoleamine 2,3-dehydrogenase (IDO) and COX2 and downregulation of TGF-β and HGF [80].

Existing research recognizes the critical role played by tissue macrophages in chronic inflammation in the elderly patients [81]. Normally, MSC exosomes support specific anti-inflammatory phenotype of regulatory macrophages by downregulating IL-22 and IL-23 [82]. MSCs directing differentiation of macrophages into the M2 phenotype via the miR-223/pKNOX1 pathway and TLR4/nuclear factor kB/phosphatidylinositol 3 kinase (PI3K)/Akt signaling cascade [83].

SASP factors modulate M1/M2 macrophage selection by shifting M2 cells towards M1 [84]. Aging MSCs enhance myeloid cell generation, innate immunity activation and affect macrophage polarization [85]. Macrophages co-cultured with young MSCs expressed M2 Arg1 and IL-10 markers, while M1-associated TNF-α was increased in cells with senescent MSCs. In addition, co-cultivation of macrophages with senescent cells increased migration capacity, which is typical for activated M1 macrophages. The MSCs released inducers of M1 differentiation, incl. IFN-γ, IL-1 and DAMP, activating the NF-κB signal [86]. Apoptosis also has a significant role in the activation of the inflammatory regulatory capabilities of MSCs [87]. Experimental studies have shown that apoptotic MSC bodies also induce polarization of macrophages towards the M2 phenotype [88]. Consequently, MSCs in an aged microenvironment have shown to lose their unique capability of M1 suppression.

The aging and accumulation of senescent cells are obviously related to cell niche, as well as to the age-related characteristics and comorbidities of the patient. While senescent MSCs have a predominantly pro-inflammatory effect, the presence of nonspecific inflammation focus stimulates cellular senescence in turn. Direct evidence highlighting the influence of MSCs inflammatory microenvironment on their aging are limited. BM-MSCs in patients with systemic autoimmune inflammation (rheumatoid arthritis), compared with MSCs obtained from healthy individuals matched by sex and age, had manifested signs of senescence and aging incl. impaired clonogenicity, decreased proliferation, and shortening of telomeres without significant differences in differentiation potential and profiles of surface markers [89]. Previous similar studies confirmed a significantly decreased proliferation and differentiation as well as chondrogenic and adipogenic potential of MSCs in cases with severe osteoarthritis [90]. Re-stimulation of MSCs from the dental pulp also induced an inflammatory response and an early onset of cellular senescence [91].

Another important class of SASP components consist of matrix metalloproteinases (MMPs). MMPs modulate the integrity of the stem cell niche by degrading the protein components of the extracellular matrix (ECM). As a result, both MSCs growth and immune cell migration are affected [92].

MSCs can differ significantly in their ability to realize an immunomodulatory effect via the secretome. Changes in miRNAs levels in extracellular vesicles (miR-223-5p, miR-127-3p and miR-125b-5p and others) explain further decrease in the immunosuppressive activity of senescent cells [93,94]. Associations between age-related changes in microRNA levels in extracellular vesicles and immunomodulatory properties of MSCs were confirmed by earlier data from rodent studies [95,96,97]. Taken together, these observations indicate that changes in the composition of MSCs EVs in senescent cells may represent a non-canonical part of SASP, which contributes to decrease in the immunomodulatory ability of MSCs. However, the effect of miRNA effects of EV MSCs on immune cells and on the original MSCs themselves, needs further investigation.

The mechanism of the immunoregulatory MSC action is primarily due to the peculiarities of intercellular interaction and paracrine signaling through the MSC secretome. Such key players of SASP as IGFBP3, IGFBP4, and IGFBP7 are suggested to be involved in mediating senescence by paracrine signaling. Components of MSC secretome are able to regulate the functional activity of both innate and adaptive immunity cells through DNA, mRNA, circular RNA (cirRNA) and long non-coding RNA [98,99]. A number of data allow us to conclude the contribution of MSC secretome to tolerogenicity. Thus, extracellular MSC vesicles inhibit the maturation and functioning of dendritic cells and T-cells due to the expression of anti-inflammatory IL-10 and TGF-b, reducing the production of pro-inflammatory IL-6 and IL-12p70 [100,101]. Moreover, MSC exosomes promote regulatory T cell differentiation by increasing IL-10 and TGF-b expression and decreasing IL-6 production and inducing mouse tolerogenic dendritic cells with low expression of costimulatory markers [102].

However, it was shown MSCs also promote T cell proliferation and influence the cell cycle through the p27kip1/CDK2 pathway [103]. Through the PI3K/Akt signaling pathway, extracellular MSC vesicles can also prevent B-cell activation [104]. Since MSCs have a high degree of immunological plasticity, it is obvious the composition of their secretome can vary greatly.

Inflammatory microenvironment also largely determines the MSC secretome composition and their interaction with immune cells (Figure 2). Thus, in acute inflammation, high concentrations of pro-inflammatory cytokines (IL-1, IL-6, TNF-α, IFN-γ) and low levels of anti-inflammatory cytokines (TGF-β, IL-10) contribute to triggering the immunosuppressive activity of MSCs (MSC2 phenotype). Cells of the MSC2 phenotype secrete enormous amounts of IDO and anti-inflammatory cytokines (TGF-β, IL-10), exerting an inhibitory effect on the activity of macrophages, neutrophils and lymphocytes. In the long-term, chronic inflammation, in contrast, MSCs acquire an immune activation phenotype (MSC1). Persistently low and comparable levels of pro-inflammatory and anti-inflammatory cytokines lead to the production of low levels of IDO and pro-inflammatory cytokines (TNF-α, IFN-γ), respectively. However, the amount of IDO secreted by MSC1 cells was found significantly lower than the amount produced by MSC2 cells during acute inflammation. Being limited to the reduced levels, the amount of IDO is inadequate to suppress the immune cells, and pro-inflammatory state overall persists.

MSC2 are capable of exerting an immunosuppressive effect due to the high production of IDO and a pool of anti-inflammatory cytokines [Figure 2]. IDO is known as an enzyme catalyzing the breakdown of tryptophan to kynurenine. An increase in IDO expression leads to the accumulation of cytotoxic kynurenine. Kynurenine inactivates immune cells and leads to suppressed immune responses. On the contrary, in chronic inflammation, MSCs-1 tend to activate immune cells [105]. In case of a chronic inflammatory response, the levels of pro- and anti-inflammatory cytokines in the MSCs microenvironment are various with subsequently different effects on MSC functions (Figure 2). While the baseline level of pro-inflammatory cytokines triggers IDO production by MSCs-1, albeit in small amounts, anti-inflammatory cytokines induce moderate secretion of pro-inflammatory cytokines via the NF-κB signaling cascade and also suppress IDO production [105,106]. Pro-inflammatory cytokines produced by MSCs may act in an autocrine manner, further enhancing limited IDO secretion. In general, a secretory combination with low levels of IDO and high levels of pro-inflammatory cytokines is not sufficient to generate an immunosuppressive response from MSCs-1. Instead, an unrestricted activation of immune cells exacerbates the inflammatory microenvironment. These mechanisms have been confirmed by IFN-γ and TNF-α titration studies in mouse, rat, and human MSCs [107,108].

In addition to pro- or anti-inflammatory microenvironmental stimuli, the acquisition of MSCs in the MSC2 or MSC1 phenotype can be induced by other exogenous factors. MSCs are able to respond through Toll-like receptors (TLRs). Short-term exposure to low levels of lipopolysaccharide (LPS) or TLR4 ligand, is able to polarize MSCs towards the MSCs-1 phenotype. LPS priming has been reported to trigger MSC secretion of pro-inflammatory cytokines (incl. IL-1β, IL-6, IL-8, IFN-γ, and TNF-α) and activate T cells [109]. According to the experimental data, the stimulation of TLR MSCs in mice led to the production of similar pro-inflammatory cytokines [105,110,111]. In contrast, exposure of TLR3 ligand to polycytidylic acid promoted the shift for MSC2 phenotype [105]. The MSC2 phenotype secretome primarily contained anti-inflammatory mediators (CC-motif chemokine ligands 10 and 5-CCL10 and CCL5, respectively) and readily suppressed T-cell activation in vitro.

## 6. Diagnostic and Therapeutic Outlooks

Currently SASP is considered as more than a secondary senescence marker, but also a clinically relevant prognosticator for various age-related diseases incl. cancer, neurodegeneration and joint demolitions [112]. Furthermore, several publications strongly proposed the cell-free MSC-secretome as an upcoming curative active substance for age-related diseases [113,114]. Since the composition of the MSC secretome can be engineered, it could be also proposed as part of a nanomedical delivery system, allowed to target dysfunctional cells or tissues specifically [113,115]. Secretom-based nanomedicine is expected to expand the therapeutic horizons for immune-mediated inflammatory diseases and conditions associated with inflammaging [116]. Despite the promising findings, those proposals suffer from the fact that relevant diagnostic panels or secretome-based test systems were not yet developed. Standardized secretome isolation, purification, and storage protocols, as well as validation tests are indispensable for clinical investigations. Furthermore, the dose and route of secretome administration must be clearly determined [14].

Clinical application of senescent cell secretome could be a dangerous major drawback of regenerative therapy for elderly patients. Similar risk is hidden in possible application of senescence MSCs as a part of biomedical product or tissue engineered graft. Their secretome was recognized as an important predictor of MSCs utility for regenerative therapy [113]. Therefore, current efforts have to be focused on issues of safety and quality control.

## 7. Discussion

Growing body of literature recognizes the secretome as a main actor in immunomodulation, tissue remodeling and homeostasis. A broad spectrum of MSC-secretome-related biological effects has been proven. It includes modulation of pro- and anti-inflammatory activity, anti-apoptotic and immunomodulatory properties. Those findings allow us to consider the secretome as a possible factor of immune homeostasis. According to the recent clinical and experimental data biocompatible secretome is far less immunogenic then MSCs but exerts similar biological effects. Thus, non-aged MSCs secretome could become an upcoming advanced tool for cell-free therapy.

However, despite the pool of emerging data supporting the proposal of MSCs secretome as a promising biomedical product, reliable studies still did not cover the gap of knowledge on secretome composition and half-life of its components [14]. Known differences in its composition depend on cell source. Thus, we can expect a novel cell-free product based on the secretome of immortalized primary-like human MSCs lines from different tissues with subsequently predicted activities [117]. Cell pre-conditioning in bioreactors has a potential to modulate their secretome and enhance therapeutic efficiency [118].

MSCs possess a high immunoplasticity and the composition of their secretome varies depending on the context. So, MSCs are capable of secreting more immunomodulatory factors under hypoxic conditions in cell-culture chamber or bioreactor [14]. In general, every microenvironment change leads to a secretome composition.

Differentiation, aging and senescence of MSCs were shown to be mediated by cellular cross-talks via growth factors and inflammatory cytokines [119].

Senescent MSCs release reactive oxygen species (ROS), causing damaging oxidative stress in the microenvironment, which contributes to the intensity of aging processes and translates senescent state to neighboring cells [92].

Observed data allow us to suggest the likely influence of inflammatory microenvironment on the processes of cell senescence. Furthermore, the hypothesis of inflammatory microenvironmental triggers for MSCs senescence and aging now received additional support. However, this hypothesis could strike the most common “gold-standard” paradigm of MSCs immunosuppressive activity. It is likely that the effects of inflammation on the MSCs and their immunomodulatory responses and secretion are determined by expression and status of the inflammation. Thus, in the acute inflammation stage with overdosed cell signaling, MSCs are able to acquire an immunosuppressive phenotype, “MSCs-2” in a similar way with other innate immune cells-macrophages [105].

The main pathways, intracellular cascades and environmental factors responsible for the senescence cells formation and tissue aging most notably are determined. However, reliable data on mechanisms of senescence cells accumulation and progression of inflammaging-related diseases is still limited. However, the phenomenon of the cells transitioning to a state of senescence and their accumulation is practically unexplored at present in the context of “host senescent cells accumulation–microbiome–immune system” interactions. The role and significance of the gut microbiota in determining the immune system cells activity, the development of inflammation processes and its possible changing during life has been studied actively in recent years. Over the past decade, research on the human microbiome has increased markedly, and large population-based studies such as the Human Microbiome Project Consortium (2012) and the Human Intestinal Metagenome Project (MetaHIT, 2010), the American Gut Project (2018), the MetaCardis project (metagenomics in cardiometabolic diseases, 2012–2018), Horizon (2020), the project of intestinal microbiome studying in the urban and rural population of the Russian Federation (2013) have significantly increased the amount of data available on the composition and function of the human microbiome. The above-mentioned data allow us to study the presence of associations between the human body and the microbiome, their relationship with the development and progression of various diseases [120,121].

Based on already known data, there is a relationship between the gut microbiome and cellular senescence, so that the metabolic profile of the gut microbiota can directly affect the accumulation of senescent cells with the SASP phenotype that can greatly disturb the homeostasis and affect the immunocompetent cell function. On the other hand, bioactive substances formed during the metabolism and fermentation of food, such as SCFA, phenols, neurotransmitters, hormones, endotoxins, ammonia, etc., can enter the systemic circulation and indirectly affect different cell types [2], including MSCs which allows us to consider the gut microbiome as a possible component and regulator of the cell senescence and aging mechanisms development.

Although the transition to the senescence state follows several rules, its natural development may be mediated by the ability of cells to counteract oxidative damage mediated by ROS and the ability to stress response [122]. Thus, experiments have shown that increased oxidative stress in experimental animal models can directly affect the increasing accumulation of senescent cells [123], while the use of antioxidants can significantly reduce cellular senescence both in vitro and in vivo [124,125]. It is known that various metabolites of the gut microbiome and probiotic bacteria or fermentation products have a strong anti-inflammatory and antioxidant effect, which can affect the regulation of the transition and accumulation of senescent cells, as well as the development of pro-inflammatory conditions, including chronic ones [2].

One of the more significant conclusions to emerge from our review is that “key drivers of inflammaging” are unacceptable sources for regenerative therapy. The relevance of MSCs secretome for determination of senescent conditions is clearly supported by the most recent findings. Thus, a precise test of the MSCs secretome would be an essential tool in addition to flow cytometry profiling prior to cell therapeutic application. We believe that our review could pave the way for upcoming studies focused on development of relevant senescent diagnostic panels for the needs of regenerative medicine.

## Figures and Tables

**Figure 1 ijms-24-06372-f001:**
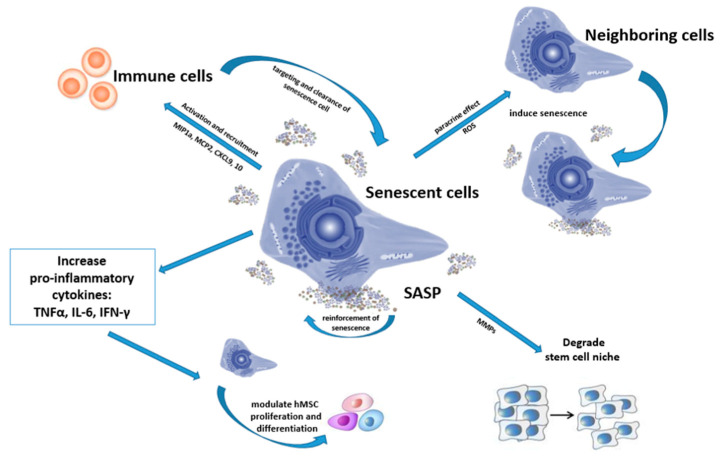
SASP effects on MSCs and their microenvironment.

**Figure 2 ijms-24-06372-f002:**
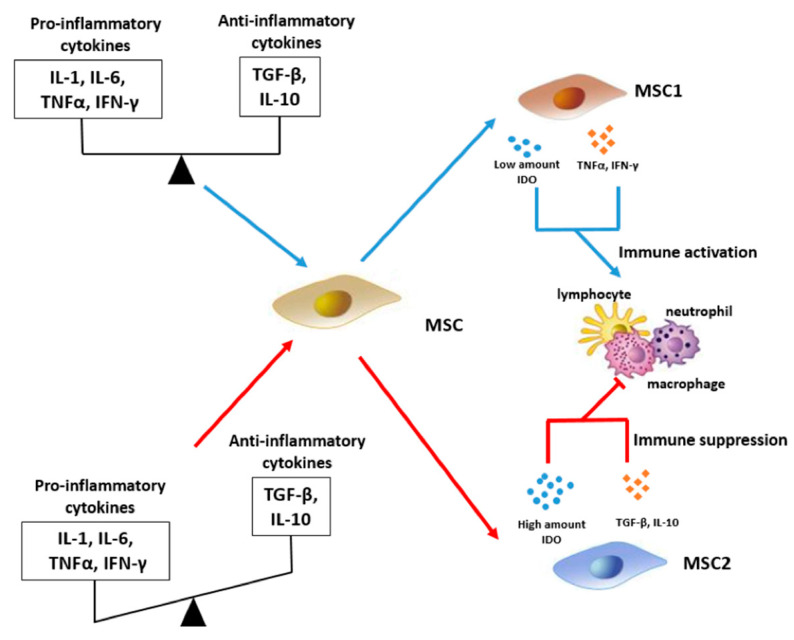
The Opposing Effects of Inflammation on the Immunomodulatory Function of hMSCs. Abbreviations: IDO, indoleamine 2,3-dioxygenase; IL, interleukin; INF, interferon; TGF, tumor growth factor; TNF, tumor necrosis factor.

## Data Availability

Data sharing not applicable. No new data were created or analyzed in this study. Data sharing is not applicable to this article.

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
