# Peer review of "Mesenchymal Stromal Cells as a Driver of Inflammaging"

_ijms, 2023, doi:10.3390/ijms24076372_

Round 1

Reviewer 1 Report

The Authors revised the literature on a hot topic argument: the immunomodulatory effects of the MSCs-secretome during cell senescence. Moreover, they provide insights suggesting the potenti al use of MSC secretome to treat different diseases associated to inflammation and aging. This is a comprehensive and useful review on this topic.

Author Response

Authors are thankful to the reviewer for the provided comment.

Reviewer 2 Report

1. Please provide more schematics.

Author Response

Authors made an essential English editing and added the graphical abstract.

Reviewer 3 Report

The review of Svetlana Lyamina and colleagues is focused on Mesenchymal Stem Cells (MSCs) as driver of inflammaging.

The take home message is not clear. Although the introduced information is of relevance on its field, the lector gets lost in the manuscript as the information does not have a proper structure, and a clear take-home-message is missing. I understand authors talk about senescence and MSCs, but not which is the information they want to transmit, reason why the manuscript should be rejected.

The manuscript presents contradictions and the lector does not understand if the point of the manuscript is to recommend or not the use of the secretome of aged MSCs. Or if it is better to stimulate them or not.

Also, authors mention inflammaging but a proper definition of inflammaging is missing.

In addition, the discussion section does not have sense, as three paragraphs about the microbiome are introduced, but the lector does not understand why are they in this section.

I would strongly encourage the authors to state a clear take-home-message: Do you want to transmit that the secretome of aged MSCs should be administered but with caution? Do you want to transmit how the senescent tissues influence MSCs? Do you want to transmit the therapeutic properties of the MSCs secretome in senescent tissues?

Once specified the the take-home-message, made a proper structure. For example, if you want to focus on MSCs and inflammaging, I propose something similar to this structure, which should be completed according to the take-home-message you want to communicate:

-          What is senescence: life expectation, senescence definition (proliferation, senescent cells, inflammaging), senescence mechanisms (SASP..)

-          Mesenchymal Stem cells: what are MSCs, characterization, paracrine mechanisms (secretome and EVs), types of MSCs, therapeutic potential and diagnostic of MSCs, aged MSCs, MSCs in a senescent microenvironment.

I persuade authors to use the information they provide, as it is correct, but with a clear take-home-message and structure.

Author Response

1. Authors are thankful to the reviewer for the provided comment. As mentioned, the article is targeted both, to assume the role of MSCs secretome in inflammaging and describe the outlooks of its application as a diagnostic and therapeutic tool. Thus, authors agreed with the Reviewer and stated an essential ‘take-home-message’ in the text as follows

Side effects and complications of MSC-based treatments increased interest in the MSCs secretome as an alternative concept for validation tests in regenerative medicine. The most recent data also proposed it as an ideal tool for cell-free regenerative therapy and tissue engineering. However, senescent MSCs secretome was shown to hold the role of ‘key-driver’ in inflammaging. 

2.  The aim of the review was defined as following:

In our review we aimed to discuss the recent evidence of associations between MSCs secretome and age related changes in tissue homeostasis via the paradigm of inflammaging. We also reviewed the relevance of secretome-based tests as an upcoming diagnostic tool for regenerative medicine.

Authors specified the overall recommendations in the concluding remarks:

One of the more significant conclusions to emerge from our review is that “key drivers of inflammaging” are unacceptable source for regenerative therapy. The relevance of MSCs secretome for determination of senescent conditions is clearly supported by the most recent findings. Thus, a precise test of the MSCs secretome would be an essential tool in addition to flow cytometry profiling prior to cell therapeutic application. We believe that our review could pave the way for upcoming studies focused on development of relevant senescent diagnostic panels for the needs of regenerative medicine.

3. Authors added the definition of inflammaging in the article:

Among the various definitions of inflammaging, the most accepted one characterizes it as ‘chronic, sterile, low-grade inflammation’ that accompanies aging as disturbances in sophisticated balance between pro- and anti-inflammatory cell signaling and responses

4. Authors agreed with the  recommendation and completely removed the microbiome  part from the article

5. Authors provided a clear take-home-message in the concluding remarks describing the dual role of MSCs secretome and providing overall recommendations for diagnostic and clinical applications.

6. Authors already devoted a part of the introduction to the basic definitions, characterization of MSCs and senescence mechanisms. However, in accordance with the recommendations, the authors concretized the definition of secretome, restructured material and reorganized the article by adding the separate paragraph named ‘Diagnostic and therapeutic outlooks’  accumulating and discussing the most recent clinical proposals and limitations of senescent MSCs & their secretome.

.

Reviewer 4 Report

This review addresses mesenchymal stromal cells as a driver of inflammation with a focus on pro-inflammatory cytokines. This is overall comprehensive research and will interest the large audience of IJMS.

Author Response

Authors are thankful to the reviewer for the provided comment. 

Authors made an essential English editing and added the graphical abstract.

Reviewer 5 Report

The authors have presented a thorough review on the role of mesenchymal tromal cells in inflammaging. There are a few minor typos but overall it is an excellent review. Also, at line 231, the abbreviations should be moved to footnotes.

Author Response

Authors are thankful to the reviewer for the provided comment, have made an essential editing and added graphical abstract.